

# Genome-wide identification and expression profile analysis of the *Hsp20* gene family in Barley (*Hordeum vulgare* L.)

Jie Li[1] and Xinhao Liu[2]

[1] College of Agronomy, Xinyang Agriculture and Forestry University, Xinyang, Henan Province, China
[2] Kaifeng Agriculture and Forestry Science Institute, Kaifeng, Henan Province, China

## ABSTRACT

In plants, heat shock proteins (Hsps) play important roles in response to diverse stresses. Hsp20 is the major family of Hsps, but their role remains poorly understood in barley (*Hordeum vulgare* L.). To reveal the mechanisms of barley Hsp20s (*HvHsp20s*) response to stress conditions, we performed a comprehensive genome-wide analysis of the *HvHsp20* gene family using bioinformatics-based methods. In total, 38 putative *HvHsp20s* were identified in barley and grouped into four subfamilies (C, CP, PX, and MT) based on predicted subcellular localization and their phylogenetic relationships. A sequence analysis indicated that most *HvHsp20* genes have no intron or one with a relatively short length. In addition, the same group of HvHsp20 proteins in the phylogenetic tree shared similar gene structure and motifs, indicating that they were highly conserved and might have similar function. Based on RNA-seq data analysis, we showed that the transcript levels of *HvHsp20* genes could be induced largely by abiotic and biotic stresses such as heat, salt, and powdery mildew. Three *HvHsp20* genes, *HORVU7Hr1G036540*, *HORVU7Hr1G036470*, and *HORVU3Hr1G007500*, were up-regulated under biotic and abiotic stresses, suggesting their potential roles in mediating the response of barley plants to environment stresses. These results provide valuable information for further understanding the complex mechanisms of *HvHsp20* gene family in barley.

## INTRODUCTION

Plants are exposed to a variety of biotic and abiotic stresses which cause disruption of protein homeostasis (*Nakajima & Suzuki, 2013*). Maintenance in functional native conformations of proteins is important for cell survival under stress. Heat shock proteins (Hsps) exist widely in prokaryotic and eukaryotic cells, are induced by many stresses (such as hot temperatures, drought, cold and various pathogen attacks, etc.), are essential components contributing to cellular homeostasis under stress conditions (*Wang et al., 2004*; *Park & Seo, 2015*).

Many Hsps are located in the cytoplasm and nucleus, and others are located in plastids, mitochondria, chloroplasts, endoplasmic reticulum or peroxisomes, suggesting they play different roles in protein homeostasis (*Scharf, Siddique & Vierling, 2001*). Based on

Corresponding author
Xinhao Liu, lxhlxhlj12@126.com

protein molecular weight and sequence homology, the plant Hsps are classified into five major families: Hsp100, Hsp90, Hsp70, Hsp60, and small Hsps (sHsps) with molecular sizes ranging from 15 to 42 kDa (*Wang et al., 2004*). Most sHsps molecular weights range from 15 to 22 kDa and they are also called Hsp20 family.

Hsp20s function as molecular chaperones by binding to partially folded or denatured proteins to prevent proteins from irreversible aggregation and keep them stable (*Eyles & Gierasch, 2010*; *Sun, Van Montagu & Verbruggen, 2002*). Hsp20s possess a conserved sequence of 80–100 amino acid residues called α-crystallin domain (ACD) in the C-terminal region (*Scharf, Siddique & Vierling, 2001*). The highly conserved ACD is flanked by a variable N-terminal domain and a short C-terminal extension. The three different regions have different functions. The N-terminal region participates in substrate binding, the ACD is involved in substrate interactions, while the C-terminus extension is responsible for homo-oligomerization (*Giese & Vierling, 2004*; *Jaya, Garcia & Vierling, 2009*). This ACD domain is comprised by two compact hydrophobic β-strand structures that are separated by α-helical region of variable length (*Bondino, Valle & Have, 2012*). However, not all Hsp20s that contain an ACD domain, and so named ACD proteins, belonged to Hsp20s because certain ACD proteins are known to have different functions (*Bondino, Valle & Have, 2012*). In plants, Hsp20s are encoded by nuclear genes. Based on the subcellular localization, sequence homology and function, Hsp20s are divided into various subfamilies (CI–CVI, MTI, MTII, ER, CP, and PX). CI–CVI subfamilies localize to the cytoplasm/nucleus, MTI and MTII subfamilies localize to mitochondria, ER, CP, and PX localize to the endoplasmic reticulum, chloroplast, and peroxisome, respectively (*Waters, 2013*). To date, the *Hsp20* gene families have been reported in many plants, such as 13 *Hsp20* genes were identified in Arabidopsis (*Scharf, Siddique & Vierling, 2001*), 23 *Hsp20* genes in rice (*Sarkar, Kim & Grover, 2009*), and 163 *Hsp20* candidates in wheat (*Muthusamy et al., 2017*). Furthermore, the biologic function of Hsp20s in protecting plants under various stress conditions is well documented in several plants including soybean (*Lopes-Caitar et al., 2013*), tomato (*Yu et al., 2016*), and pepper (*Guo et al., 2015*).

The genomic analysis of barley *Hsp20* family has been conducted by *Reddy et al. (2014)* and identified 20 *sHsp* genes (*Reddy et al., 2014*). Last year, a high-quality reference genome assembly for barley was presented, which provides a convenience for further understanding the barley *Hsp20* gene family (*Mascher et al., 2017*). In the present study, 38 *HvHsp20* genes were identified using the new barley genome by bioinformatics methods. Then the analysis of identified *HvHsp20* genes in the sequence features, chromosomal locations, phylogenetic relationships, and dynamic expression patterns in response to biotic and abiotic stress conditions were conducted, which provided valuable information for further investigations of the barley *Hsp20* gene family.

## MATERIALS AND METHODS

### Genome-wide identification of Hsp20 proteins and chromosomal location

The barley genome sequences were downloaded from Ensembl database (http://plants.ensembl.org). The Hsp20 candidates were identified as the methods of

*Lozano et al. (2015)*. The HMMER software (http://hmmer.janelia.org) was used to build the predicted barley protein data. The Hidden Markov model (HMM) profile of Hsp20 (PF00011) was downloaded from the Pfam protein family database (http://pfam.xfam.org/) (*Finn, Clements & Eddy, 2011*) for the identification of Hsp20 proteins. The obtained high-quality protein set ($E$-value $< 1 \times 10^{-10}$) was aligned and used to construct a barley specific Hsp20 HMM using hmmbuild from the HMMER v3 suite. The new barley-specific HMM was used to select all proteins that were downloaded from the barley genome (Hordeum_vulgare.IBSC_v2) with an $E$-value $< 0.01$. The selected barley Hsp20 sequences were matched to Hsp20 conserved domains (PF00011) using ClustalW (*Larkin et al., 2007*), then removed the incomplete sequences. The remaining Hsp20 protein sequences were submitted to Pfam (http://pfam.xfam.org/) and the Simple Modular Architecture Research Tool (SMART, http://smart.embl-heidelberg.de/) to confirm the conserved Hsp20 domain.

The chromosomal positions of the *HvHsp20* genes were acquired from Ensembl database (Hordeum_vulgare.Hv_IBSC_PGSB_v2.39). Each of these genes was mapped on chromosomes using the Map Gene2chromosome (MG2C, version 2.0) tool (http://mg2c.iask.in/).

## Conserved motifs and transcript structures analysis

The conserved motifs among subgroups of Hsp20 proteins were analyzed by the program of Multiple Expectation for Motif Elicitation (MEME; version 4.10.0) with the following parameters: maximum of 10 motifs, any number of repetitions, and an optimum motif width of 6–50 amino acid residues. Exon–intron structures of barley Hsp20 genes were identified on the Gene Structure Display Server (GSDS, http://gsds.cbi.-pku.edu.cn/) (*Hu et al., 2014*).

## Phylogenetic analysis

To illuminate evolutionary relationship of Hsp20s, the genomic sequence information of Arabidopsis, rice, and sorghum was downloaded from the Ensembl Plants database (http://plants.ensembl.org/index.html). All of the acquired sequences were aligned using ClustalW (version 2.1) program with the default parameters (*Larkin et al., 2007*). The phylogenetic tree was constructed by MEGA X with bootstrap test of 1,000 times, which was visualized in Figtree (http://tree.bio.ed.ac.uk/software/figtree/).

## Cis-acting regulatory element prediction in promoter regions

The upstream sequences (1.5 kb) of *HvHsp20* sequences were submitted to the PlantCARE database (http://bioinformatics.psb.ugent.be/webtools/plantcare/html/) to computationally predict various regulatory elements based on positional matrices, consensus sequences, and individual sites on particular promoter sequences (*Lescot et al., 2002*), such as abscisic acid-responsive elements, heat stress elements, low-temp and salt stresses, TC-rich repeats, and W boxes, etc.

## Expression analysis of *HSP20* genes under biotic and abiotic stresses

The Illumina RNA-seq data (GSE117068, GSE101304, and GSE82134) were downloaded from the NCBI Sequence Read Archive database to study the expression patterns of

*HvHsp20* genes under biotic and abiotic stress conditions. Digital gene expression analysis of the identified *HvHsp20* genes was investigated using Hisat2, Htseq, and DESeq2 (*Wen, 2017*). Expression ratios of treated samples relative to control that met the threshold criteria of $|logFC| > 2.0$ with a $P < 0.05$ was considered as differently expressed.

# RESULTS

## Identification of Hsp20 family members in barley

To identify all the Hsp20 candidate members in barley, we searched for sequences that contained the ACD domain using the HMMER with barley-specific HMM model of PF00011, and a total of 143 Hsp20s were identified. After removing the repetitive and/or incomplete sequences, the rest of Hsp20s were submitted to Pfam (http://pfam.xfam.org/) and SMART (http://smart.embl-heidelberg.de/) to confirm the ACD domain. Finally, 96 candidate Hsp20 protein sequences were confirmed in barley, and were coded by 38 *Hsp20* gene sequences. The protein and gene sequences of barley Hsp20s appear in Table S1.

## Phylogenetic analysis of *HvHsp20* gene family

To analyze the phylogenetic relationships of *Hsp20* genes in barley, rice, sorghum, and Arabidopsis, a phylogenetic tree was constructed using full-length Hsp20 proteins (Fig. 1). All these Hsp20s were grouped into seven distinct subfamilies based on their predicted subcellular localizations and the number of ACD domain, including C (cytosol), MT (mitochondria), CP (chloroplast), ER (endoplasmic reticulum), PX (peroxisome), and ACD subfamilies followed with 13 Hsp20s could not be classified into any subfamilies. In addition, the C subfamily was subdivided into 4 subgroups (CI–CIV) and MT subfamily was subdivided into two subgroups (MTI and MTII). In barley, C subfamily has a maximum of 32 HvHsp20s followed by CP subfamily consisting of 25 HvHsp20s. Most of Hsp20s were classified into C subfamily, suggesting that cytosol might be a main functional area for plant Hsp20s. We found an interesting result that CP and MT subfamily members had a close relationship (Fig. 1), which is consistent with the opinion that MT subfamily evolved later from the CP subfamily (*Waters, 2013*).

## Gene structures and motifs of HvHsp20s

The GSDS website was used to analyze the structural characteristics of the *Hsp20* genes in barley. The structural characteristic of 86 *HvHsp20* genes that have normal intron length is shown in Fig. 2A. For another 10 *HvHsp20* genes, the length of intron is much longer than that of extron, was shown in Fig. 2B. The number of introns in the *HvHsp20* genes varies from 0 to 10, with 35 *HvHsp20* genes (36%) were intronless. About 45% (43) of *HvHsp20* genes contain one intron, and 18 (19%) had two or more introns. One *HvHsp20* gene contains 10 introns and two contain six introns (Figs. 2A and 2B). According to the number of introns, these *HvHsp20* genes were divided into three patterns: the first pattern has no intron, the second pattern has one intron, and the third pattern has more than one intron (*Ouyang et al., 2009*). Most of *HvHsp20* genes (81%, 78) belonged to the first and the second pattern, which was similar to the results reported on Hsp20 proteins in tomato (*Yu et al., 2016*).

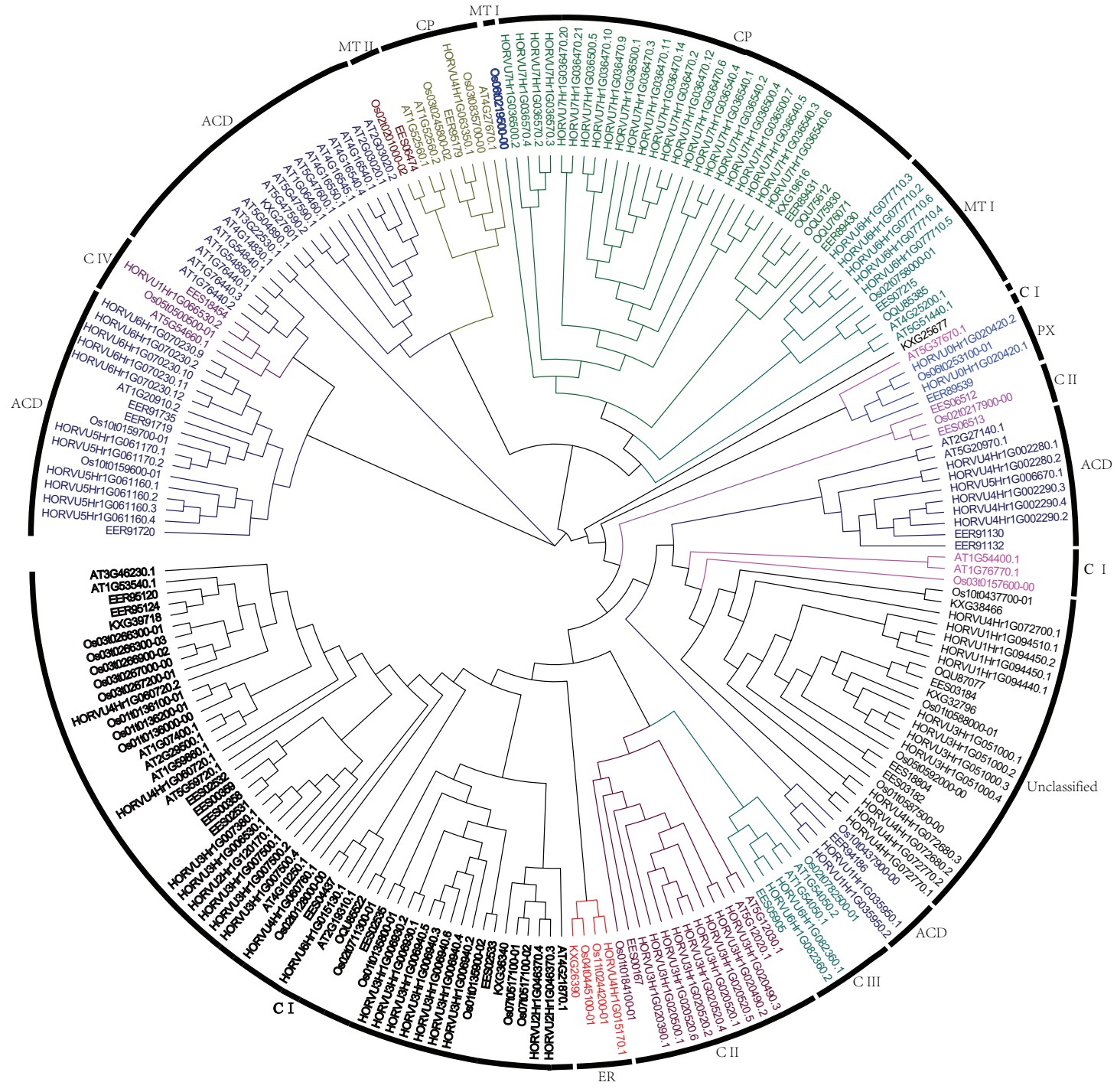

**Figure 1 Phylogenetic relationship of Hsp20s from Arabidopsis, rice, sorghum, and barley.** The phylogenetic tree was constructed using MEGA X with 1,000 bootstrap replications.

Using the MEME tool, 10 types of consensus motifs in HvHsp20 proteins were identified (Fig. 3). The lengths of these conserved motifs varied from 15 to 50 amino acids (Motif logos shown in Fig. S1). The majority of the HvHsp20 proteins (50) contained
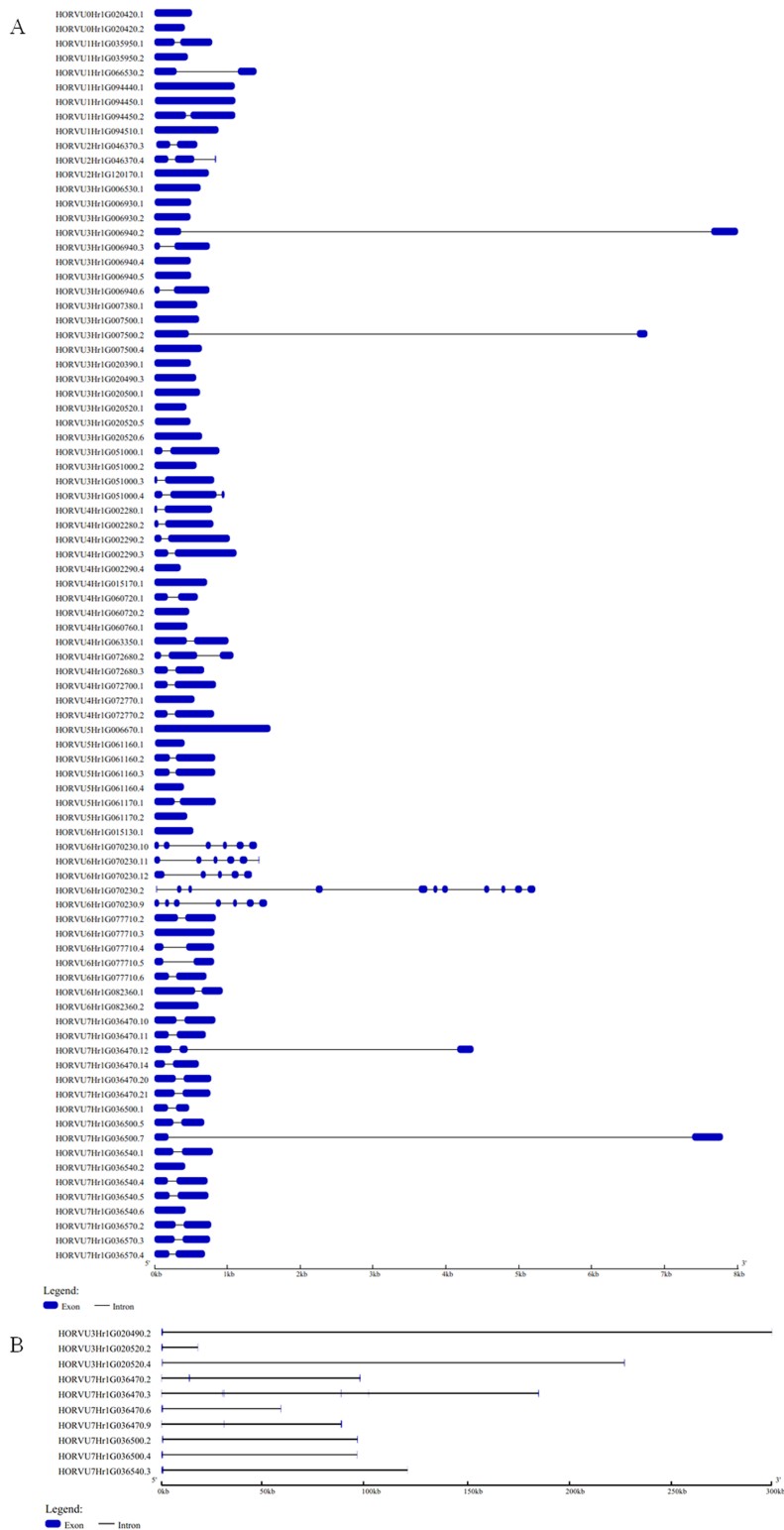

**Figure 2 Exon/intron structures of barley *HvHsp20* genes.** (A) The gene structures of 86 *HvHsp20* that have normal intron length. (B) The structural characteristic of 10 *HvHsp20* genes that possess too long introns. Boxes filled with blue represent exons, solid black lines represent introns. Scale at bottom is in kb.

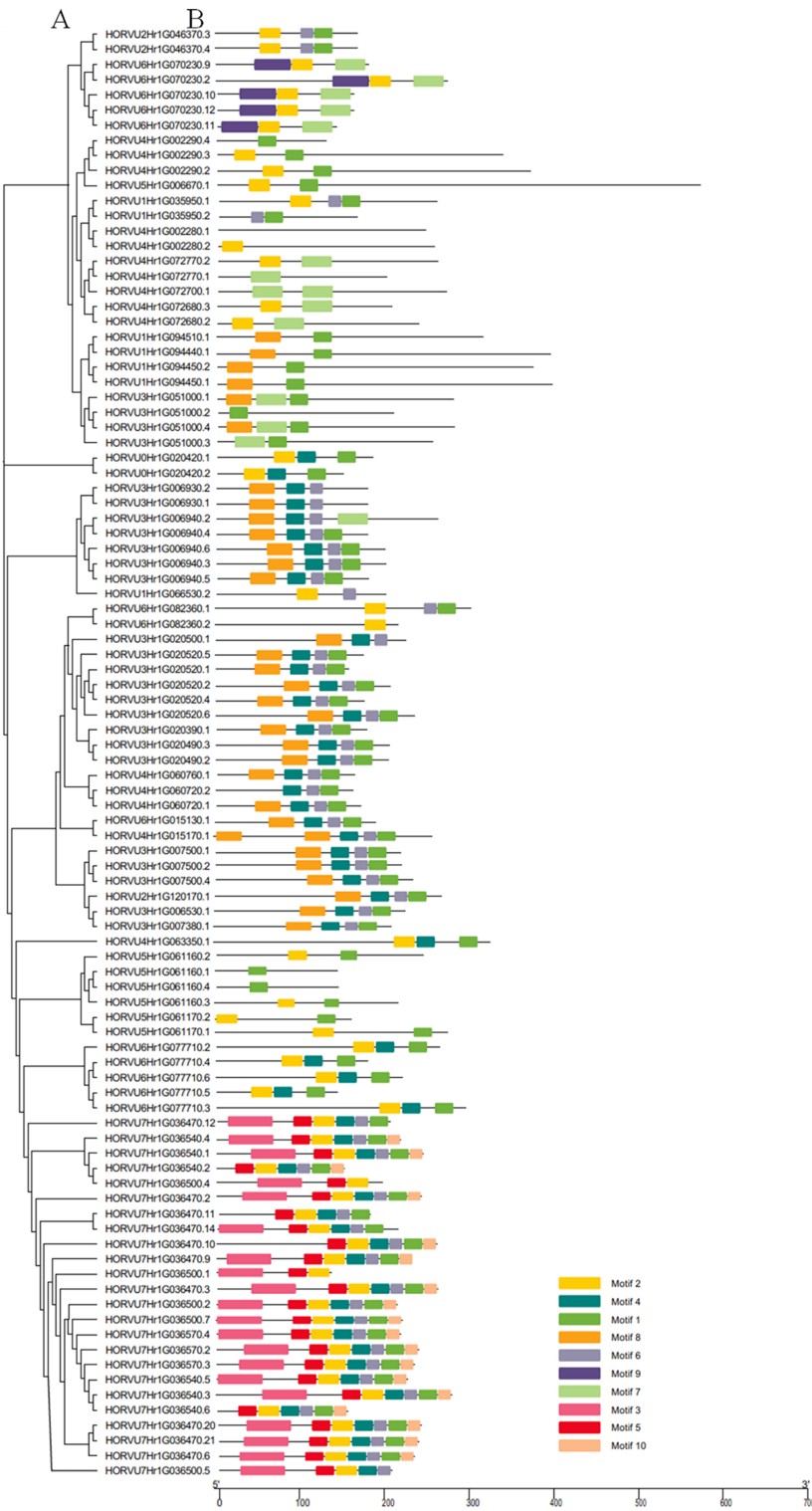

**Figure 3 Phylogenetic relationships and motif compositions of HvHsp20 proteins.** (A) Multiple alignment of Hsp20 domain was executed using ClustalW and the phylogenetic tree was constructed with MEGA X using full-length amino acid sequences of identified Hsp20 proteins in barley. (B) Distributions of conserved motifs in HvHsp20 proteins. Each colored box represents a motif in the protein.

Motif 1 while Motif 9 was found in only 5 HvHsp20 proteins. The multiple sequence alignment analysis and sequence logo revealed that the ACD domain was formed by two conserved regions, conserved region I and conserved region II (Fig. 4), moreover, most of motif 1 and 6 located in consensus region I, and consensus region II contained motif 5, 7, and 8. Motif 3 and 9 located in the N-terminal of Hsp20, motif 10 located in C-terminal, while motif 4 and 2 located between consensus region I and consensus region II of ACD domain (Table S2). Furthermore, HvHsp20 proteins in the same subgroup present similar patterns of motif distribution, suggesting that these genes might have relatively high conservation.

## Chromosomal location of *HvHsp20* genes

Among 38 predicted *HvHsp20* genes, 37 are randomly distributed across the seven barley chromosomes, except for the *HORVU0Hr1G020420* which could not be located at anyone chromosome (Fig. 5). The distribution of the *HvHsp20* genes on each chromosome is uneven. Some *HvHsp20* genes are clustered on chromosomes 3 and 7, some are scattered on chromosomes 2 and 6, and most of these *HvHsp20* genes were located at the distal ends of the chromosomes. The number of *HvHsp20* genes on each chromosome is different. Chromosome 3 has the most number of *HvHsp20* genes (10 genes), followed by chromosomes 4 (nine genes). Chromosomes 6 and 7 have the same number of *HvHsp20* genes (four genes). Chromosome 1 has 5 *HvHsp20* genes, chromosome 5 has three genes, while only two genes on chromosome 2.

## Stress-related *cis*-elements in the *HvHsp20* promoters

To further explore the regulatory mechanisms of *HvHsp20* genes in response to stress conditions, the *cis*-elements in the promoter region (the 1.5 kb upstream sequences from the translation start sites) of the *HvHsp20* genes were further analyzed. Nine stress response elements were analyzed and displayed in Fig. 6, including ABRE, DRE_core, G-Box, LTR, TC-rich repeats, W-box, MBS, CCAAT-box, and MYB. Among these 38 genes, except for 2 *HvHsp20* genes (HORVU6Hr1G015130 and HORVU7Hr1G036470), the other possessed at least 1 type of stress-response-related *cis*-element; and HORVU1Hr1G035950, had eight categories *cis*-elements, indicating that the expressions of *HvHsp20* genes were related to various stress responses. In total, one or more G-Box existed in 29 *HvHsp20s* (76%), followed by ABRE which was found in 28 *HvHsp20s* (74%). In addition, one or two TC-rich repeats existed in 12 *HvHsp20s*, while DRE_core, W-box, LTR, MBS, CCAAT-box, and MYB were presented in 20, 16, 14, 18, 11, and 5 *HvHsp20s*, respectively (Table S3). The analysis of these cis-elements suggested that *HvHsp20* genes could response to different stress conditions.

## Expression profiles of *HvHsp20* genes under biotic and abiotic stresses

To explore the responses of *HvHsp20* genes to various stresses, expression patterns of *HvHsp20* genes in response to abiotic stresses (arsenate treatment and heat) and biotic stress (powdery mildew infection) were investigated using RNA-seq data.



**Figure 4 Multiple sequence alignment of ACDs of HvHsp20s.** Names of all members are listed on the left side of the figure. Conserved amino acid residues are indicated by gray background. Two conserved regions (conserved region I and conserved region II) are underlined at the bottom.

Peer J

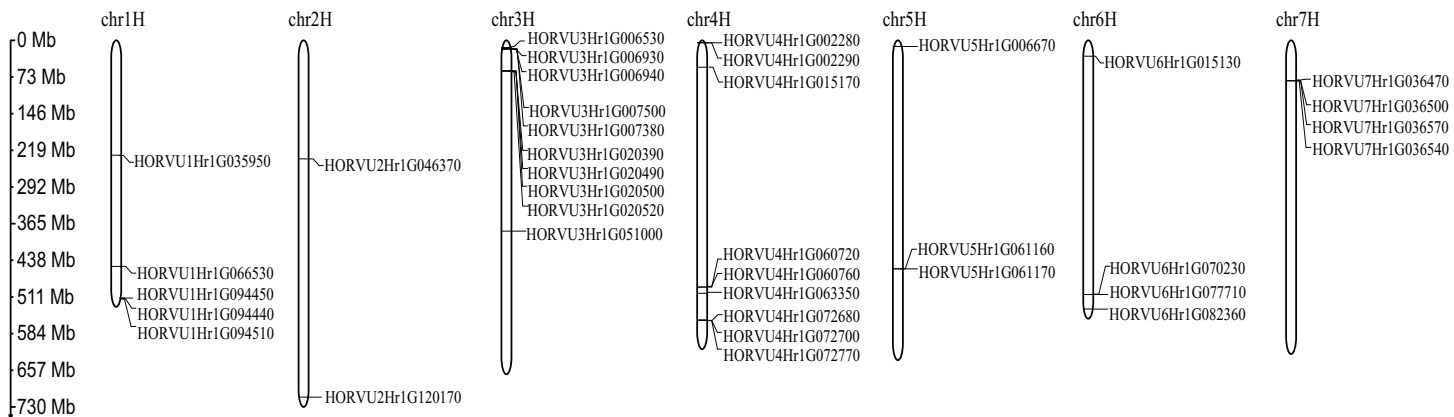

**Figure 5 Chromosome localizations of barley *HvHsp20* genes.** Chromosome numbers are shown on top of each bar. Scale on left is in Mb.

Overall, the 37 out of the identified 38 *HvHsp20* genes showed differential expression patterns under these conditions (Fig. 7). Arsenate stress induced relatively more fluctuations in the transcript abundance of *HvHsp20* than that of the heat stress and powdery mildew infection. A total of seven *HvHsp20* genes were induced in response to arsenate stress in arsenate sensitive barley genotype (ZDB475), and 19 genes increased in arsenate and phosphate stress (Fig. 7B). In arsenate tolerant barley genotype (ZDB160), 19 *HvHsp20* genes showed up-regulated expression under both arsenate stress and arsenate and phosphate stress (Fig. 7A). Some increased expression in *HvHsp20* genes induced by arsenate stress displayed a slight declined expression trend in arsenate and phosphate stress. In addition, more up-regulated *HvHsp20* genes in ZDB160 than in ZDB475 implied *Hsp20s* played critical roles in response to arsenate stress. Over 60% of *HvHsp20* genes were up-regulated under abiotic stresses, which went up to 90% in heat stress (Figs. 7C and 7D), while for powdery mildew stress, more than 80% of *HvHsp20* genes were down-regulated (Fig. 7E). In addition, three *HvHsp20* genes, HORVU7Hr1G036540, HORVU7Hr1G036470, and HORVU3Hr1G007500, were up-regulated under biotic and abiotic stresses (Fig. 7), suggesting they have major functions in response to various stress conditions. In total of 10 *HvHsp20* genes showed differential expression trends in all of the stress conditions, three *HvHsp20* genes (HORVU3Hr1G006930, HORVU4Hr1G072700, and HORVU6Hr1G070230) showed specific-expression under biotic stress (Fig. S2).

## DISCUSSION

Hsp20 proteins play vital roles in plant growth and development processes as well as biotic and abiotic stress responses. With the availability of the whole genome sequence, the *Hsp20* family genes have been identified in many plants, such as Arabidopsis and rice (*Scharf, Siddique & Vierling, 2001*; *Sarkar, Kim & Grover, 2009*). Preliminary analysis of the barley *Hsp20* gene family has been performed by *Reddy et al. (2014)*. Benefiting from the high-quality reference genome information of barley (*Mascher et al., 2017*), by constructing barley-specific HMM, we identified 38 *HvHsp20* genes coding 96 HvHsp20

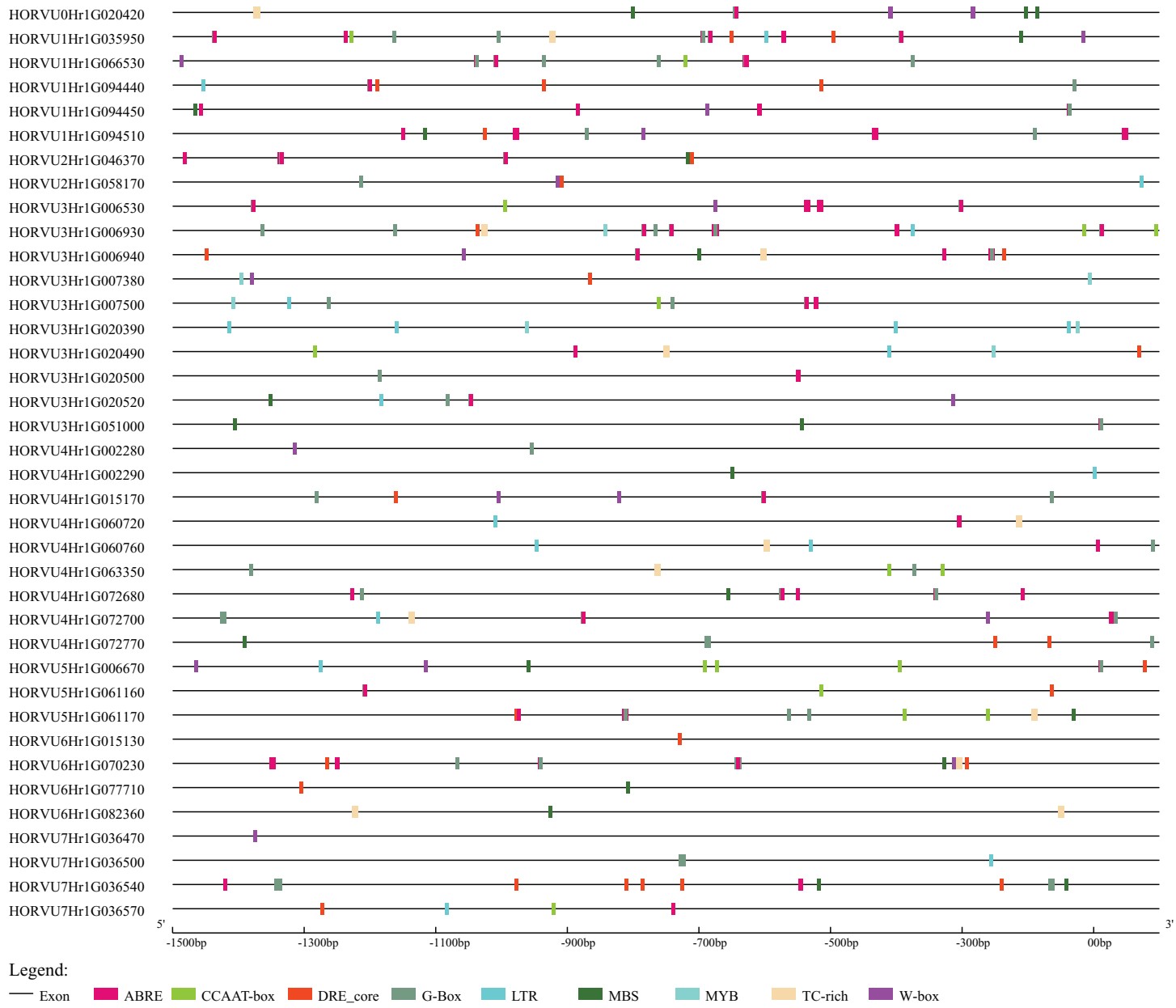

**Figure 6 Predicted *cis*-elements in *HvHsp20* promoter region.** Promoter sequences (−1.5 kb) are analyzed by PlantCARE.

proteins. The number of identified *HvHsp20* genes was more than *Reddy et al. (2014)* reported, which might be because we took different identification methods and/or more *Hsp20* genes were annotated recently. Then we analyzed their structure, chromosomal location, phylogeny, and expression pattern diversity with respect to biotic and abiotic stresses, which provides useful information for understanding the *HvHsp20* gene family and will establish a basis for future analyzing functional divergence of the *Hsp20* genes in barley.

The number of identified *Hsp20* genes in graminaceous plants, such as rice and wheat was 39 and 163, respectively (*Ouyang et al., 2009*; *Muthusamy et al., 2017*).
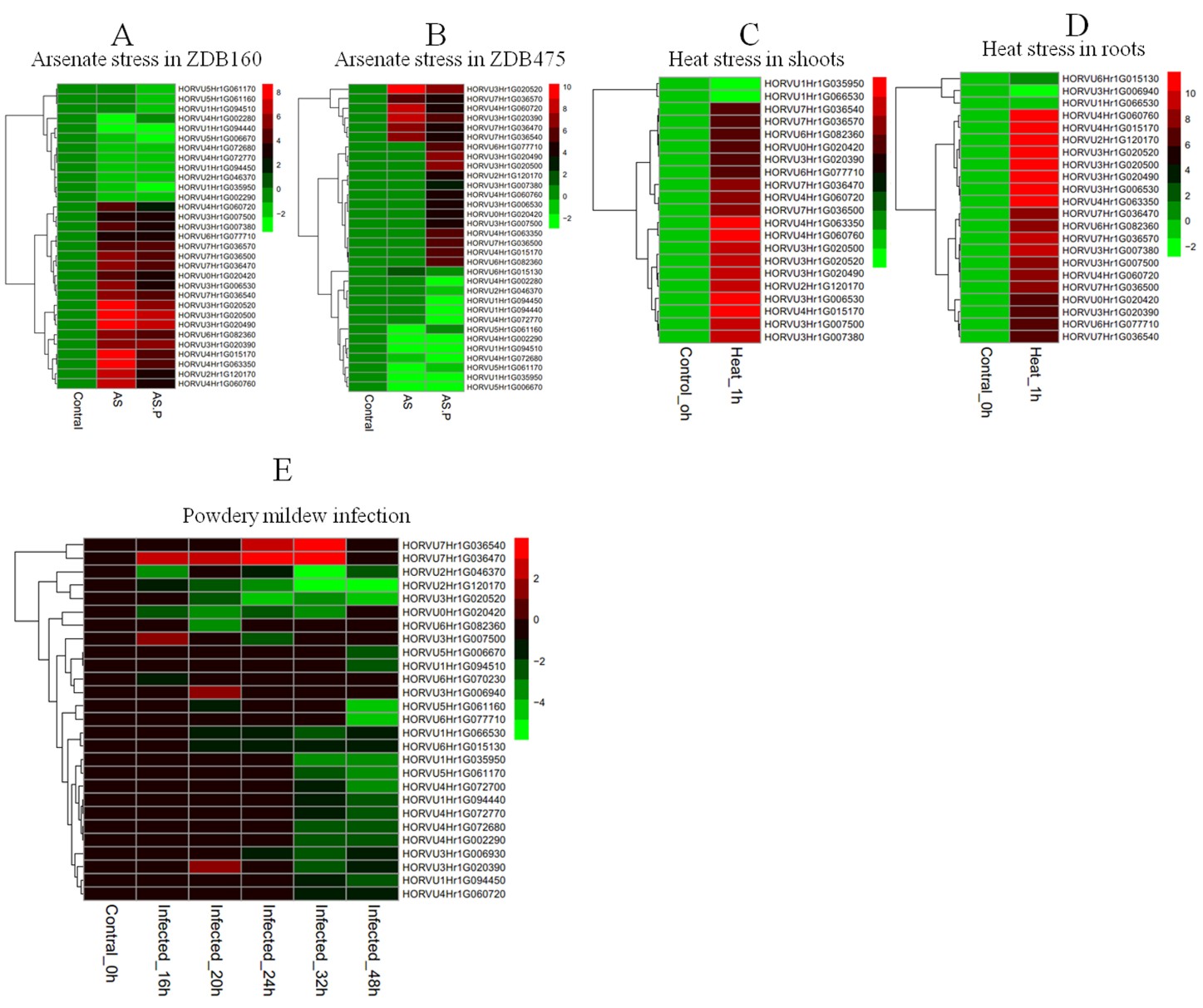

**Figure 7 Expression profiles of *HvHsp20* genes under various biotic and abiotic stresses.** Green blocks represent decreased and red blocks represent increased transcript levels relative to the respective control. (A) Expression profiles of identified *HvHsp20* genes under arsenic stress in tolerant barley genotype (ZDB160). (B) Expression profiles of *HvHsp20* genes in arsenate sensitive barley genotype (ZDB475) under arsenic stress. (C) Expression profiles of *HvHsp20* genes in barley shoots under heat stress. (D) Expression profiles of *HvHsp20* genes in barley roots under heat stress. (E) Expression profiles of *HvHsp20* genes in barley leaves during powdery mildew infection.               

The number of members in barley was 38, which was close to the number of members in rice, but was fewer than in wheat, suggesting the possibility of a gene gain event from diploid to hexaploid during the evolutionary process. Previously, plants Hsp20 family members were classified into five subfamilies according to their subcellular localization including C, CP, MT, ER, and PX subfamilies, and ACD subfamily (*Scharf, Siddique & Vierling, 2001*; *Siddique et al., 2008*). In our study, to reveal phylogenetic relationship of Hsp20 family members, Hsp20s from Arabidopsis, rice, sorghum, and barley, were used to

construct a phylogenetic tree (Fig. 1). The results showed that these Hsp20s existed in seven subfamilies including some Hsp20s that could not be grouped. Interestingly, we found that 25 barley Hsp20s belonged to CP subfamily, which was much more than the number of Hsp20s in rice and wheat (*Ouyang et al., 2009*; *Muthusamy et al., 2017*), suggesting barley Hsp20s might have undergone duplication and/or recombination events during evolution.

The exon/intron structure plays a vital role in organismal evolution (*Xu et al., 2012*). Here, we found the positions, sizes, and sequences of the introns were quite different between the identified *HvHsp20* genes (Fig. 2). About 36% *HvHsp20* genes do not have introns, which is slightly lower than in rice genes predicted to be intronless (48.72%) (*Ouyang et al., 2009*), most of these *HvHsp20* genes belonged to CI and CII subfamilies. Many members of the CP and MTI subfamilies had only one intron, and the length of introns is relatively short. This is consistent with the report that plants tend to retain the genes with no intron or a short intron (*Mattick & Gagen, 2001*). To respond to environmental challenges, most genes with fewer introns are rapidly activated (*Jeffares, Penkett & Bähler, 2008*). *Hsp20* genes are one of the rapidly expressed genes under a variety of stresses (*Sarkar, Kim & Grover, 2009*; *Yu et al., 2016*). The absence of introns or their presence with smaller size may in accordance with the needs of the rapid induction of *Hsp20* genes. In this study, we found that most of the *HvHsp20* genes with one or no intron were induced under several stresses, but genes with many introns, such as HORVU6Hr1G070230, were induced during only powdery mildew infection. Furthermore, the majority members in the same phylogenetic subfamily had similar motif compositions (Fig. 3). This correlation between intron numbers and motif arrangement supported the previous classification of the *HvHsp20* genes. These results may provide facilitation to identify the functions of *HvHsp20* genes and further to discover their function in responses to environmental stresses.

From the expression analysis, it has been demonstrated that *Hsp20* genes play important roles in the control of plants in response to diverse environmental challenges (*Waters, 2013*). In this study, we investigated the expression profiles of the *HvHsp20* genes in barley after biotic and abiotic stress treatments (Fig. 7). The data demonstrated that numerous *HvHsp20* genes were significantly induced to a larger extent under abiotic stresses including arsenic and heat stress, and 21 *Hsp20* genes showed differential expression patterns under two stress treatments, which belonged to CI and CII subfamilies, confirming the protective role of *HvHsp20* family members in barley. For heat stress, among differential expressed *HvHsp20* genes, the vast majority of them grouped in CI, CII, and CP subfamilies were up-regulated both in barley shoots and roots. Previous research showed that the CI and CII subfamilies both have chaperone activity, and the CI proteins are lower efficiency than CII members in preventing irreversible aggregation (*Basha et al., 2010*). However, by knockdown the CI and CII genes, other studies suggested that the CI sHSPs have thermoprotective roles but the CII sHSPs do not (*Port et al., 2004*; *Tripp, Mishra & Scharf, 2010*). The biochemical and biological differences between the subfamilies may be consistent with the patterns of N-terminal

sequence conservation within subfamilies (*Waters, 2013*). Continued studies of the relationship between sequence differences and functional changes of *Hsp20* genes will be crucial to further understanding the function of sHSPs. Furthermore, we found that the arsenic stress inducibility of *HvHsp20* genes in tolerant plants was stronger than that in susceptible plants, suggesting that *HvHsp20* genes play vital roles in response to heavy metal stress.

Several Hsp20 proteins also participate in the interaction of plants and pathogens, such as virus, bacteria and fungus (*Park & Seo, 2015*). In the present study, five *HvHsp20* genes showed an increased expression during the first few hours of powdery mildew infection, then the expression dropped rapidly following the infection. The results are consistent with the report that biotic stress can induce the gene expression of some but not all sHSPs (*Siddique et al., 2008*), suggesting that *HvHsp20* genes may confer biotic stress tolerance in barley. Among these five *HvHsp20* genes, three genes (HORVU7Hr1G036540, HORVU7Hr1G036470, and HORVU3Hr1G007500) were also significantly induced under abiotic stresses including arsenic and heat stress, suggesting these three *HvHsp20* genes might play important roles in mediating the response of barley plants to various stresses. However, further analysis is needed to investigate how the three *HvHsp20* genes influence plant defense response.

# CONCLUSIONS

In the current study, a genome-wide analysis of barley *HvHsp20* genes family was performed, and 38 putative *HvHsp20* genes were identified. A comprehensive analysis of *HvHsp20* genes on gene structures, chromosomal location, phylogenetic relationship stress-related cis-elements, and expression patterns under biotic and abiotic stresses, were conducted by using bioinformatics and RNA-seq data. Most *HvHsp20* genes belonging to CI, CII, and CP subfamilies showed differential expression under stress conditions, indicating that *HvHsp20* genes play important roles in response to stress. This study provided comprehensive information on the *HvHsp20* gene family in barley and will aid in investigating the function of *HvHsp20* genes.

### Funding

This work was supported by the Youth Foundation of Xinyang Agriculture and Forestry University (No. 201701006) and Henan Provincial South Henan Crop Pest Green Prevention and Control Academician Workstation. The funders had no role in study design, data collection and analysis, decision to publish, or preparation of the manuscript.

### Grant Disclosures

The following grant information was disclosed by the authors:
Youth Foundation of Xinyang Agriculture and Forestry University: 201701006.
Henan Provincial South Henan Crop Pest Green Prevention and Control Academician Workstation.
## Competing Interests

The authors declare that they have no competing interests.

## Author Contributions

- Jie Li conceived and designed the experiments, prepared figures and/or tables, authored or reviewed drafts of the paper, approved the final draft.
- Xinhao Liu performed the experiments, analyzed the data, prepared figures and/or tables, approved the final draft.

## Data Availability

The raw data is available in the Supplemental Files.

## Supplemental Information

Supplemental information for this article can be found online at http://dx.doi.org/10.7717/peerj.6832#supplemental-information.

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
