# Peer review of "Genome-wide identification and expression profile analysis of the Hsp20 gene family in Barley (Hordeum vulgare L.)"

_PeerJ, doi:10.7717/peerj.6832_

## Round 0.1 · original submission · Minor Revisions

Dear author

Your paper has been assessed by two reviewers and myself as academic Editor.

As you could see below, the manuscript was rated positively by all reviewers, with several minor revisions suggested. I congratulate you for the nice piece of work, which will add value to PeerJ.

Please address some of the concerns of the reviewers and sumbit a revised version of the manuscript. Please include a response to each reviewer. I have been informed that PeerJ is now offering a language correction service for a fee. Please consider it before submitting the new version of the manuscript.

Reviewer 1 ·

Basic reporting

No comment

Experimental design

No comment

Validity of the findings

No comment

Additional comments

This manuscript presents identification of barley Hsp20 gene family in Arabidopsis, rice and sorghum based on presence of an ACD domain. Using phylogenetic analysis, authors report four subfamilies of HSP20 multiple gene family. Authors describe features of coding sequence, chromosomal location, intron-exon contain of HSP20 gene family and use RNA-seq data from NCBI-SRA database to reveal up- and down regulation of barley HSP20 genes upon biotic and abiotic stress. I feel this manuscript will be interest to many researchers in the field. However, there are some issues as follows needs to be addressed.

1) Line 141-147, authors identified 38 Hsp20 gene sequences after removing the repetitive sequences. In my opinion authors should check whether incomplete full-length sequence could be completed using the chromosome sequence information, and amended the amino acid sequence displayed in Table S1.

2) line 151-155, authors introduce a classification of identified HSP20s into C, MT (MTI and MTII), CP, ER, PX and ACD subfamilies. Giving that classification is based in protein location, could it determined by signal peptide or a signature?. In my opinion authors should add information, at some extend, to help audience to distinguish hsp20 into C to ACD subfamilies based in amino acid sequences. Are there reasons to include multiple protein coding gene models for a HSPs in the Fig 1?. For instance, HORVU5Hr1G061160.1, HORVU5Hr1G061160.2, HORVU5Hr1G061160.3 and HORVU5Hr1G061160.4 HSP proteins at the ACD subfamily.

3) Some typos in the manuscript. For instance, line 95 “MATHERIALS”, line 115 “misfits”, line117 “identified”, and line 289 “Hsp20”.

4) The Fig 2 displayed 86 gene models that is less than the 96 reported by authors.

5) In my opinion, it should be useful for the audience if the authors add a little bit on the 10 motifs reported (Fig 3) regarding to ACD domain, N-terminal and C-terminal extension. Do motifs fall into or outside ACD domain?

5) The Fig 4 displayed 35 HSP20 genes, is there a reason for the missing HSP20 gene in the figure?

6) Line 126, Section “Cis-acting regulatory…”, It should be useful to explain to the audience features of the cis-element detection such as strand and matrix similarity for the presence/absence of cis-element.

7) Fig 6, panel B, the scale for the expression levels is missing at “Heat stress in shoots”

·

Basic reporting

The language is clear and used properly in my opinion, the references are sufficient although could be more clear in the information being cited. For example line 66 Liberek citation is mostly about HSP 70 and 100 and although mentions about sHSP the information from this could be obtain from other publications that would focus on sHSP from plants.
The structure is well define, however in the abstract it should be corrected the sentence that mentions that this information is needed to reveal the mechanism..... Since the data presented in this manuscript is mostly descriptive and it does deal with deciphering any of the mechanism of action of these proteins.

Experimental design

The research is well done and proper, probably syntenic analysis could be made to understand the evolution of the families of HSP20 proteins found.

Validity of the findings

The findings are expected but the information is required to obtain more information about possible target proteins to use for heat stress survival in future experiments. The RNA seq data is acceptable, there are always more conditions to be tried and time points of expression. But for the title of the work it seems appropriate. The conclusions are concrete and accurate with regard to the data and link to the research question from this manuscript

Additional comments

From the phylogeny data it should be taken into account the different genome duplication between the different plants choose, a synteny approach my provide more information about the evolution of this family. The novel signatures for this family that are presented correspond to any other family of proteins? this should be mention.

---

## Round 0.2 · accepted · Accept

Dear authors

The reviewer has confirmed that you have addressed all the minor corrections and that the manuscript is now aceptable.

I congratulate you for the nice piece of work, which will add value to PeerJ.

# Reviewer 1 ·

Basic reporting

Pass

Experimental design

Pass

Validity of the findings

Pass